# Voluntary Physical Exercise Reduces Motor Dysfunction and Hampers Tumor Cell Proliferation in a Mouse Model of Glioma

**DOI:** 10.3390/ijerph17165667

**Published:** 2020-08-05

**Authors:** Elena Tantillo, Antonella Colistra, Laura Baroncelli, Mario Costa, Matteo Caleo, Eleonora Vannini

**Affiliations:** 1Neuroscience Institute, National Research Council (CNR), 56124 Pisa, Italy; elena.tantillo@sns.it (E.T.); anto_colistra@yahoo.it (A.C.); baroncelli@in.cnr.it (L.B.); costa@in.cnr.it (M.C.); caleo@in.cnr.it (M.C.); 2Classe di Scienze, Scuola Normale Superiore, 56124 Pisa, Italy; 3IRCCS Fondazione Stella Maris, 56128 Calambrone, Italy; 4Department of Biomedical Sciences, University of Padua, 35121 Padua, Italy; 5Fondazione Umberto Veronesi, 20122 Milan, Italy

**Keywords:** glioma, voluntary physical exercise, tumor proliferation, motor cortex, GL261, motor tests

## Abstract

Currently, high-grade gliomas are the most difficult brain cancers to treat and all the approved experimental treatments do not offer long-term benefits regarding symptom improvement. Epidemiological studies indicate that exercise decreases the risk of brain cancer mortality, but a direct relationship between physical exercise and glioma progression has not been established so far. Here, we exploited a mouse model of high-grade glioma to directly test the impact of voluntary physical exercise on the tumor proliferation and motor capabilities of affected animals. We report that exposing symptomatic, glioma-bearing mice to running wheels (i) reduced the proliferation rate of tumors implanted in the motor cortex and (ii) delayed glioma-induced motor dysfunction. Thus, voluntary physical exercise might represent a supportive intervention that complements existing neuro-oncologic therapies, contributing to the preservation of functional motor ability and counteracting the detrimental effects of glioma on behavioral output.

## 1. Introduction

Voluntary physical exercise is among the components of the well-known paradigm of environmental enrichment (EE), a very effective strategy to restore plasticity and elicit recovery from different neurodevelopmental disorders [1,2,3,4,5,6,7]. EE is known to profoundly affect the central nervous system (CNS) at the functional, anatomical, and molecular levels, both during the critical period and adulthood [8,9]. EE consists of rearing animals in large social groups and in the presence of running wheels and many stimulating objects that are regularly changed and substituted to stimulate explorative behavior, curiosity, and attentional processes [9,10]. Recent studies focusing on the visual system have shown that these effects are associated with the recruitment of previously unsuspected neural plasticity processes [11]. In the early stages of brain development, EE triggers a marked acceleration in the maturation of the visual system, with maternal behavior acting as a fundamental mediator of the enriched experience in both the fetus and the newborn [9]. In the adult brain, EE enhances plasticity in the cerebral cortex, allowing for the recovery of visual functions in amblyopic animals [6]. The molecular substrate of the effects of EE on brain plasticity is multi-factorial, with reduced intracerebral inhibition, enhanced neurotrophin expression, and epigenetic changes at the level of the chromatin structure [11]. These findings shed new light on the potential of EE as a non-invasive strategy to ameliorate deficits in both CNS development and in the treatment of neurological disorders. In particular, a recent study has demonstrated that glioma-bearing mice exposed to EE showed a reduced tumor size with respect to a standard environment, together with an increase of interleukin 15 (IL-15) and Brain-derived neurotrophic factor (BDNF) intracerebral levels [12].

Although EE is composed of physical exercise, social interactions, cognitive stimulation, and visual enrichment, it has been proven that the beneficial effect of EE is principally due to voluntary physical exercise, carried out through running wheels [6,13]. Interestingly, experiments on rodents have proved that both EE and voluntary physical exercise represent a preventive, beneficial therapeutical treatment for neurodegenerative diseases [14,15,16,17,18]. In particular, it has been demonstrated that exercise prevents the decline of the hippocampus from aging and ameliorates many neurodegenerative diseases by increasing adult hippocampal neurogenesis and activity-dependent synaptic plasticity. Moreover, physical exercise activates a multitude of molecular factors that promote mechanisms of brain health acting at different levels (i.e., neurotrophins modulation, chromatin remodeling, and the reduction of gamma-aminobutyric acid (GABA) [19]. Importantly, physical exercise and short-term deprivation of the amblyopic eye promoted the long-term recovery of both visual acuity and stereopsis in adult amblyopic subjects [20,21], suggesting that physical activity could be used as a non-invasive strategy to elicit plasticity in the human brain.

Glioblastoma multiforme (GBM) remains one of the most difficult cancers to treat, with a median survival of 15 to 17 months and the 5-year overall survival is less than 5% [22,23]. All the approved experimental treatments for high-grade glioma patients do not offer long-term benefits regarding symptom improvement or quality of life [24,25]. In this context, we reasoned that exercise might alter the tumor microenvironment at multiple levels and hence impact glioma progression.

Thus, the experiments reported here were designed to uncover the effects of voluntary physical exercise on tumor development and general animal well-being.

## 2. Methods

### 2.1. Animals

A total of 24 adult (age > postnatal day 60) C57BL/6J mice were used for this study. Animals were bred in our animal facility and housed in a 12 h light/dark cycle with food and water available ad libitum. All experimental procedures were performed in conformity with the European Communities Council Directive 86/609/EEC and were approved by the Italian Ministry of Health (260/2016-PR).

### 2.2. GL261 Cells and Tumor Induction

GL261 cells were grown in complete Dulbecco’s modified Eagle’s medium (DMEM) containing 10% newborn calf serum, 4.5 g/L glucose, 2 mM glutamine, 100 UI/mL penicillin, and 100 mg/mL streptomycin at 37 °C in 5% CO_2_ with media changes three times per week. Under ketamina/xylazina anesthesia (100/10 mg/kg intraperitoneally (i.p.), C57BL/6 mice received a stereotaxically guided injection of 40,000 GL261 cells (20,000 cells/1 μL Tris HCl solution) into the primary motor cortex (specifically, at the level of cortical forelimb representation (coordinates: 1.75 mm lateral and 0.5 mm anterior to the bregma) using a Hamilton syringe. The GL261 cell solution was slowly delivered at a depth of 0.9 mm from the pial surface. Body temperature was monitored with a rectal probe and maintained at 37.0 °C with a thermostated electric blanket during the surgery. An oxygen mask was placed in front of the animal mouth to facilitate breathing. To prevent dehydration, a subcutaneous injection of saline (0.9% NaCl, 1 mL) was delivered at the end of the procedure [26,27,28].

### 2.3. Running Wheels

On day 12 after the tumor induction, mice were randomized to either running or sedentary conditions. Runners were allowed to perform voluntary physical exercise by placing running wheels in their cages (Figure 1A). Each wheel was connected to an activity monitoring system that permitted calculations of the distance traveled by each animal, which was recorded every day. The group of glioma-bearing animals that performed voluntary physical exercise was called “running” and was compared with the “sedentary” mice (i.e., glioma-bearing animals that had no access to running wheels in their cages). Six naive mice were also exposed to wheels and their running was monitored, where it was found that their traveled distance was not statistically different from the running glioma-bearing mice (Figure 1B).

### 2.4. Motor Tests

Mice were tested in two different motor tasks, i.e., grip strength and grid walk. Each animal performed all the tests before the GL261 injection (baseline measurement) and 3, 5, 7, 9, 12, 15, 17, 19, and 21 days after the tumor implant [27]. To exclude any influence of circadian rhythms on behavior, all tests were performed during the same time interval each day (2:00–5:00 p.m., light phase). All the behavioral tests and analyses were performed blind to the experimental condition.

### 2.5. Grip Strength Test

The animal was placed over a base plate in front of a grasping bar (trapezoid-shaped) whose height was adjustable. The bar was fitted to a force transducer connected to a Peak Amplifier (Ugo Basile S.R.L, Varese, Italy). When pulled by the tail, the animal grasped at the bar (rodents instinctively grasp anything they can to try to stop this involuntary backward movement) until the pulling force overcame their grip strength. After the animal lost its grip on the grasping bar, the peak amplifier automatically stored the peak pull-force achieved by the forelimbs and showed it on a liquid crystal display. Three trials per day were performed for each animal and their average was calculated. Values obtained were normalized to the baseline [27]. The performances of eight sedentary and six running glioma-bearing mice were evaluated.

### 2.6. Grid Walk

Mice were placed on a grid (32 cm × 20 cm with 11 mm × 11 mm mesh), on which to walk properly, they had to put their paws on the wires; these were considered correct steps. Instead, when one of the limbs fell in a hole of the grid, a wrong step was calculated. The test took 5 min for each animal and it was filmed using a camera positioned in front of the grid. A mirror placed below the grid helped to understand which of the limbs made the mistake. The video analysis was made by using a frame by frame reproduction. Scoring was done separately for affected (i.e., contralateral to GL261 injection) and unaffected (ipsilateral) limbs [27,29]. Performance with the affected limbs (contralateral to tumor induction) was calculated as follows:affected limb wrong steps % = (affected limb wrong steps/affected limb total steps) × 100.(1)

The performances of seven sedentary and five running glioma-bearing mice were analyzed.

### 2.7. Three-Chamber Sociability Test

Eighteen days after the tumor induction, mice were tested in the three-chamber sociability task. Each mouse had a 5-min habituation period in the social test box with the doors open, and then a 10-min choice task between the chamber containing an unfamiliar mouse and the chamber containing an object. During the 5 min habituation period, in order to demonstrate that mice did not have a bias for spending time on one particular side, we monitored the exploration of both chambers. To avoid possible preferences for one of two parts, the position of the stranger mouse and object were randomized between animals. The amount of time spent exploring the mouse and the object were recorded and evaluated by the experimenter blind to the mice’s housing condition [30,31]. The arena and object were cleaned with 10% ethanol between trials to stop the build-up of olfactory cues. The performances of four sedentary and six running glioma-bearing mice were evaluated.

### 2.8. Open Field Test

The arena used was square (60 cm × 60 cm) and constructed with poly(vinyl chloride). Eighteen days after the tumor induction, each mouse’s position was continuously recorded by a video tracking system (Noldus Ethovision XT, Wageningen, the Netherlands) for 10 min in the arena. The total movement and the velocity of the animal were automatically computed [32,33]. The arena was cleaned with 10% ethanol between trials to stop the build-up of olfactory cues. The performances of four sedentary and six running glioma-bearing mice were analyzed.

### 2.9. Nesting Test

Eighteen days after the tumor induction, animals were single-housed and provided with three pieces of 60 × 50 mm^2^ tissue paper in the front part of the cage. Sixteen hours later, analysis of the built nest was done in a qualitative manner. The qualitative assessment determined the presence or absence of a proper nest and quantification of the torn paper was done by three experimenters blind to the mice’s housing condition [34,35]. The performances of four sedentary and six running glioma-bearing mice were evaluated.

### 2.10. Stereological Reconstruction of the Tumor Volume

Animals were deeply anesthetized with chloral hydrate and perfused transcardially with PBS 1×, followed by a fixative (4% paraformaldehyde, 0.1 M sodium phosphate, pH 7.4) 23 days after the tumor induction. Brains were gently removed, post-fixed for 4 h in the same fixative at 4 °C, cryoprotected via immersion in 30% sucrose with 0.01% sodium azide solution at 4 °C, and frozen by isopentane. Coronal sections (45 μm) were cut on a microtome and collected in PBS. Serial sections (one of six, spacing factor) were stained with Hoechst dye (1:500, Sigma Merck, Darmstadt, Germany) for the evaluation of the glioma volume. The tumor volume was measured in coronal sections using a Zeiss microscope (Zeiss, Oberkochen, Germany) and the Stereo Investigator software (MicroBrightField, Williston, VT, USA). In each of the selected slices, the area of the tumor was delineated with a magnification of 10×. The tumor volume was calculated by summing the measured areas and multiplying by the spacing factor and by the thickness of the slice (45 μm).

### 2.11. Immunohistochemical Analysis of Glioma Cell Proliferation

For the evaluation of glioma cell proliferation, brain serial sections were stained with two markers of proliferation, Ki67 (1:400; Abcam, Cambridge, MA, USA) and BrdU (1:500; Abcam, Cambridge, MA, USA). To label proliferating cells, BrdU (5-bromo-2′-deoxyuridine) was intraperitoneally (i.p.) administered (50 mg/kg) 24 h before sacrificing the animals. Slices were incubated with fluorophore-conjugated secondary antibodies (Jackson Immunoresearch, Ely, UK) and with Hoechst dye (1:500; Sigma Merck, Darmstadt, Germany) for nuclei visualization [28]. To improve consistency, for each cohort of mice (grafted in the same surgical session) proliferation rates and tumor volumes were normalized to the mean value obtained in control (i.e., sedentary) conditions.

### 2.12. Image Acquisition and Analysis of Tumor Cell Proliferation

Fluorescent images were acquired using a Zeiss Axio Observer microscope equipped with Zeiss AxioCam MRm camera (Carl Zeiss MicroImaging GmbH, Oberkochen, Germany). Slides were coded to ensure blinding during the acquisitions. Images were acquired through a 10× objective and joined precisely using Zen Blue Edition software 3.1 (Carl Zeiss MicroImaging GmbH, Oberkochen, Germany). For the quantification of the density of Ki67 or BrdU-positive cells, we acquired the whole area of the tumor for each coronal section. Images were processed using Fiji Image J software (National Institute of Health, Bethesda, MD, USA). The density of the proliferating cells was expressed as the fraction of area occupied by Ki67 or BrdU-positive cells relative to the total tumor area [12,28]. To minimize the variation between different immunostaining reactions and to normalize the data, we always processed a control group (i.e., sedentary) together with the running animals. Measurements of tumor proliferation were normalized to the mean of the control (sedentary) group in the same immunohistochemical reaction. Statistical comparisons were performed by cumulating the data obtained from all sections in the same experimental group.

### 2.13. Statistical Analysis

All statistical analyses were performed using the SigmaStat Software 12.0 (Systat Software Inc, San Jose, CA, USA). Differences between two groups were assessed with a two-tailed *t*-test. For longitudinal motor tests, two-way RM ANOVA was performed. The level of significance was *p* < 0.05.

## 3. Results

### 3.1. Voluntary Physical Exercise Reduced Glioma Cell Proliferation

We first tested the impact of voluntary physical exercise on glioma cell proliferation. Tumors were induced using a unilateral injection of syngeneic GL261 cells into the mouse motor cortex and 12 days after tumor induction, i.e., at symptomatic stages of the pathology [28], mice were randomized to either remain in standard cages or be placed into cages containing a running wheel (Figure 1A). Each wheel was connected to an activity monitoring system that allowed for calculating the distance traveled by each animal. We found that glioma-bearing mice ran on average 1.215 km/week, comparable to naive animals reared in the same conditions (*t*-test, *p* = 0.79; Figure 1B). To determine the impact of the running wheel access on glioma proliferation (Figure 1A), we sacrificed both running and sedentary glioma-bearing animals 23 days after tumor implantation, and we performed immunostaining to label proliferating tumor cells. Specifically, we used immunostaining to assess the density of positive glioma cells for two proliferation markers, Ki67 and 5-bromo-2′-deoxyuridine (BrdU, administered i.p. 24 h before sacrifice). Representative immunolabelling for Ki67 and BrdU is reported in Figure 1C,E. The quantifications showed that voluntary physical exercise impacted on glioma proliferation; indeed, it significantly reduced the density of both Ki67^+^ and BrdU^+^ cells within the tumor mass (*t*-test, *p* < 0.001 for Ki67 and BrdU; Figure 1D,F).

### 3.2. Comparison of Tumor Volumes in Running vs. Sedentary Glioma-Bearing Animals

To determine the impact of running wheel access on glioma size, we performed Hoechst staining to label the tumor masses [26,36]. Despite the significant reduction in tumor proliferation rate reported above, we did not find any differences in the tumor sizes of the running group relative to the sedentary mice (*t*-test, *p* = 0.167) (Figure 2).

### 3.3. Delayed Impairment of Motor Functions in Glioma-Bearing Mice That Underwent Voluntary Physical Exercise on Running Wheels

To determine the impact of exercise on glioma-induced neural dysfunction, we longitudinally followed the motor performances [27] with specific tests (i.e., grip strength and grid walk). The grip strength test (a measure of forelimb force) showed a deterioration of the performance in glioma-bearing mice until day 12 (i.e., when the running wheels were put into the cages) (Figure 3A). The onset of voluntary physical exercise slowed down the deterioration of motor functions, while sedentary glioma-bearing mice continued to worsen (Figure 3A; two-way RM ANOVA, post hoc Holm Sidak, *p* = 0.033). In particular, on day 21, we detected a significant enhancement of the force exerted by running vs. sedentary mice (*t*-test, *p* < 0.05; Figure 3B).

In the grid walk test, we noticed an initial improvement in the performance of all glioma-bearing animals, likely due to a learning phase (Figure 3C) [27]. Indeed, task practice led to a reduction in the percentage of errors (foot faults) between days 3 and 9; then, starting from day 9, the performance of the affected limbs (i.e., contralateral to the glioma cell implant) progressively decayed (Figure 3C). In the animals that were given access to the running wheels, we detected a transient, significant enhancement of performance at day 15 (two-way RM ANOVA, *p* = 0.009). Furthermore, in this motor task, running mice exhibited a delay in the deterioration of motor function (Figure 3C), which eventually caught up with the control group at day 21 (*t*-test, *p* > 0.05; Figure 3D).

### 3.4. Behavioral Evaluation of the Physical Exercise Effect on Running and Sedentary Glioma-Bearing Mice

We also performed several tests aimed at evaluating the general healthiness of the animals. We tested mice 18 days after the tumor induction, i.e., when the motor symptoms were already evident (Figure 3A,C) [27]. The nesting test showed that runner glioma mice were able to build a better nest in comparison with the sedentary group (*t*-test, *p* = 0.0034; Figure 4A). In addition, the open field test demonstrated that running glioma mice moved for a longer distance (*t*-test, *p* = 0.009; Figure 4B) and reached a higher velocity (*t*-test, *p* = 0.0107; Figure 4C) relative to the sedentary glioma mice. However, no differences were observed between the experimental groups in the mean distance from the arena center and the number of entries into the central portion of the arena (*t*-test, *p* = 0.537 and *p* = 0.26, respectively; Appendix A). We further investigated the mood state and the sociability of the animals using the three-chamber test [37] and we did not find any differences between the two experimental groups in the three parameters considered (time spent exploring the object, *t*-test, *p* = 0.39; time spent exploring the stranger, *t*-test, *p* = 0.51; time spent exploring their own part of the arena, *t*-test, *p* = 0.16; Appendix A).

## 4. Discussion

Taken together, the results presented here show that voluntary physical exercise, even when performed at a symptomatic stage of the disease, reduced the proliferation rate of tumor cells, transiently ameliorated motor function deficits associated with glioma progression, and supported a good healthiness of mice affected by glioma.

We carried out our experiments by allowing voluntary physical exercise in a cohort of mice bearing tumors in frontal areas, starting from day 12 after the glioma induction. The immunohistochemical and behavioral analyses revealed that physical exercise per se exerted a protective effect against the glioma progression (Figure 1). These data are in agreement with previous evidence, where physical exercise was added to standard chemotherapy (i.e., temozolomide treatment) and found to prolong the survival of glioma-bearing mice [38]. It is also worth mentioning that exercise is a strong independent predictor of survival in malignant recurrent glioma [39]. In particular, median survival is significantly enhanced in glioma patients reporting more than 9 metabolic equivalent (MET) h/week, equivalent to brisk walking for 30 min on 5 days/week [39]. This result highlights the possibility that some circulating factors (e.g., BDNF, IGF1), whose release is activated by running, might act to slow down glioma growth [4,12,19,40]. However, when looking at the tumoral volume of the sedentary and running glioma-bearing animals, we did not find any difference between the two groups (Figure 2); despite the beneficial circulating factors due to physical exercise, the tumoral growth was probably too fast to be significantly impacted by a mild treatment like physical exercise.

Nevertheless, using a battery of well-characterized motor tests [27,29], we evaluated whether the exercise-mediated reduction in cell proliferation could also influence motor dysfunctions induced by the progression of the glioma growth. Specifically, the grip strength test revealed a progressive deterioration of the performance of glioma-bearing mice until the beginning of physical exercise on day 12; thereafter, the performances of the running group ameliorated the deterioration relative to the sedentary group, which instead continued to decline until day 21 (Figure 3). Moreover, in the grid walk test, we detected a slightly delayed deterioration of motor activity in the glioma-bearing running vs. sedentary groups (Figure 3). We also evaluated the overall healthiness of glioma-bearing animals using the behavioral tests of open field, three chambers, and nesting. Intriguingly, we found that running glioma-bearing mice had better mobility (in terms of velocity and distance traveled; Figure 4) and a higher capability of building a nest in comparison with the sedentary group (Figure 4). We did not find any differences in the anxious state (Appendix A) nor in the sociability (Appendix A) of the two experimental groups. Our hypothesis was that, despite the glioma growth, voluntary physical exercise might help in preserving the motor capabilities of the paws, resulting in better and more coordinated movement that makes the mice healthier.

It might be argued that exposure to running wheels triggers maintenance of motor performances directly due to training, rather than via the reduction of glioma proliferation. Despite this caveat, the data on motor performances show that physical exercise allows for slowing down the glioma-induced motor dysfunction with potential impacts on the overall patients’ quality of life.

## 5. Conclusions

Taken altogether, our data support the encouraging relationship between appropriate physical exercise and the amelioration of survival outcomes in patients with neurologic malignancies [41,42,43], suggesting a protecting effect of peritumoral tissue against glioma progression [39]. At the molecular level, it is well established that exercise modulates a variety of systemic (i.e., metabolism, inflammation) and neuronal factors (e.g., neurotrophins) [44,45] that, in turn, may alter tumor signaling pathways and/or modulate the glioma microenvironment [12,40]. For instance, many studies have demonstrated that voluntary physical exercise increases IGF1 and BDNF levels [19,46], dampens oxidative stress [47,48,49], and ameliorates microcirculation [50]. The activation of all these pathways might have led to maintaining the good healthiness of the running mice affected by glioma (vs. the sedentary group), contributing to a transitory improvement of their motor abilities and slowing down their unavoidable motor deterioration.

In summary, the present data demonstrated the protective role of voluntary physical exercise in slowing down glioma proliferation, with the consequent preservation of functionality of the peritumoral tissue. Importantly, we have demonstrated that this intervention was also beneficial when administered at a symptomatic stage of the disease. Thus, even if not resolved, protocols aimed at voluntary physical exercise could be implemented in the clinical setting as adjuvant and non-invasive therapy for glioma treatment in order to ameliorate patients’ quality of life.

## Figures and Tables

**Figure 1 ijerph-17-05667-f001:**
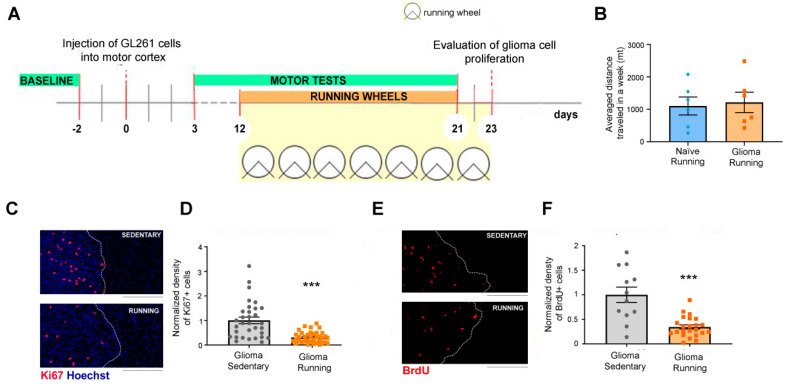
Impact of physical exercise on tumor proliferation. (**A**) Experimental protocol: mice were injected with GL261 cells, and from day 12 after tumor induction, a group was allowed to perform voluntary physical exercise (“running”) by placing a running wheel in their cages. The control group (“sedentary”) had no running wheels in their cages. Glioma-bearing animals were also tested with motor tasks before (“baseline”) and at different timings after the glioma induction in order to longitudinally monitor their motor performances. (**B**) Glioma-bearing mice traveled the same distance in kilometers as the naïve mice (*p* = 0.79). (**C**) Representation of Ki67 (red) and Hoechst staining (blue) for sedentary (SEDENTARY) and running (RUNNING) glioma-bearing animals. Note the decreased number of cells that were positive for Ki67 in the running group. Scale bar: 200 μm. (**D**) The normalized density of tumor area occupied by Ki67-positive cells in sedentary and running glioma-bearing animals. The running group had a strongly decreased proliferation of tumor cells when compared to the sedentary group. The scatterplot reports data from all the brain sections examined within each experimental group (*t*-test, *p* < 0.001). (**E**) Representation of BrdU staining for sedentary (SEDENTARY) and running (RUNNING) glioma-bearing animals. Note that runners had a decreased number of cells positive for BrdU staining. (**F**) The normalized density of tumor area occupied by BrdU positive cells in sedentary and running glioma-bearing animals. The scatterplot reports data from all the brain sections examined within each experimental group. Note the diminished number of tumoral proliferating cells for runner glioma-bearing mice (*t*-test, *p* < 0.001). Data are expressed as mean ± SEM. *** *p* < 0.001. Symbols (diamonds, squares and circles) represent animals.

**Figure 2 ijerph-17-05667-f002:**
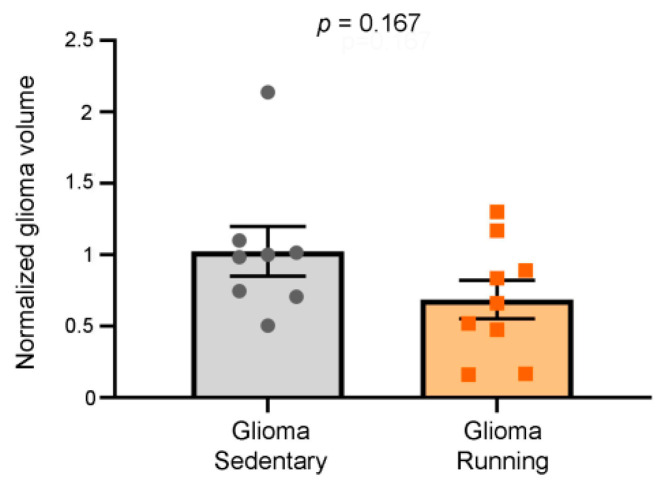
Glioma size in the running and sedentary mice. The tumor volume sizes for sedentary and running glioma-bearing mice were normalized based on the average of the sedentary group. No statistical difference was observed in the glioma size between the sedentary and running animals (*p* = 0.167). Data are expressed as mean ± SEM; symbols (squares and circles) represent animals.

**Figure 3 ijerph-17-05667-f003:**
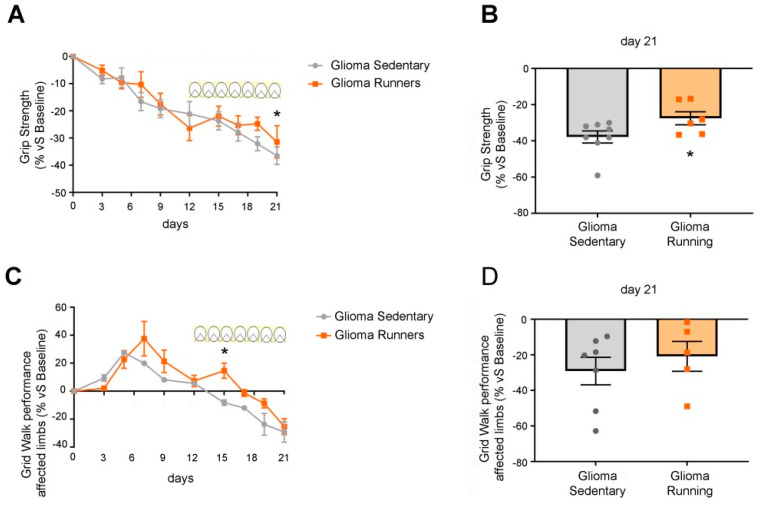
Running glioma-bearing mice showed a delayed deterioration of motor performance. (**A**,**B**) Grip strength test. The performance of running glioma-bearing mice (orange line, *n* = 6) displayed a significant difference relative to sedentary glioma-bearing mice on day 21 (grey line, *n* = 9; two-way RM ANOVA, treatment × day interaction *p* = 0.033), with better overall preservation of limb strength. (**C**,**D**) Grid walk test. A significant difference in the number of correct steps performed with affected limbs was evident between the two groups (grey line, *n* = 7 for sedentary glioma-bearing mice; orange line, *n* = 5 for running glioma-bearing animals) 15 days after the GL261 cell injection (two-way RM ANOVA on rank-transformed data, treatment × day interaction *p* < 0.05). On day 21, the grid walk performance of runners was not statistically different from the sedentary group. Data are expressed as mean ± SEM. * *p* <0.05, symbols (squares and circles) represent animals.

**Figure 4 ijerph-17-05667-f004:**
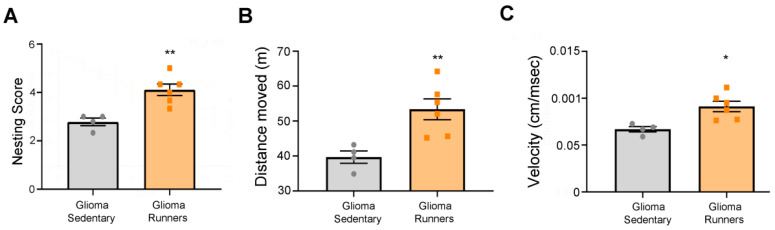
Running glioma-bearing mice showed improved motor capabilities. (**A**) Nesting test. Running glioma mice (*n* = 6) produced a better nest in comparison with the sedentary group (*n* = 4; *t*-test, *p* = 0.0034). (**B**,**C**) The open field test showed that running glioma-bearing animals (*n* = 6) travelled more ((**B**), *t*-test, *p* = 0.009) and with an increased velocity ((**C**), *t*-test, *p* = 0.0107) than sedentary (*n* = 4) glioma mice. Data are expressed as mean ± SEM. * *p* < 0.05, ** *p* < 0.01, symbols (squares and circles) represent animals.

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
