# Peer review of "Voluntary Physical Exercise Reduces Motor Dysfunction and Hampers Tumor Cell Proliferation in a Mouse Model of Glioma"

_ijerph, 2020, doi:10.3390/ijerph17165667_

Round 1

Reviewer 1 Report

Introduction

pg 1, line 29-30: though environmental enrichment may be well known to you, not all readers may be as familiar with it. Please at least list, if not summarize, the components other than voluntary physical exercise

page 2, lines 49-51: Please make explicit if these statements apply only to animal models (I assume so since it ends with running wheels).

Similarly, throughout the manuscript, please make explicit when referenced studies dealt with animal vs human subjects

pg 2, lines 68-74: this belongs in methods

pg 2, lines 74-77: this belongs in discussion or conclusion

Methods

pg 2, line 81: How many mice?

Methodological procedures seem adequately described

After reading the results, I am wondering why Mann-Whitney tests were mentioned in the statistical analysis subsection, since they weren't reported as used in the results. Were all data normally distributed with equal variances, thus not requiring Mann-Whitney tests? If so, why bother mentioning them in the methods?

Results

pg 4, lines 173-178: this is redundant of the methods

pg 4, line 179: where is the data used for t-test analysis comparing runners and naive animals coming from? There's no mention of naive animals in this study

Figure 1. What is the yellow background section with many circles that are divided by right angles? This isn't explained in the figure caption

pg 5, line 210: delete "we found a trend for decreased tumor volumes;" there was no significance, therefore no effect. Using trend in this way is incorrect in statistics. Similarly in Figure 2, delete "there is a tendency for decrease glioma size in running animals"--your statistics do not support that statement

pg 6, lines 228: which effect is the reported p value of 0.033 referring to?

pg 6, line 235: the omnibus F statistic from the RM-ANOVA would not have detected a single day's change; what is the specific p value and what effect or extra test did you specifically run to detect that day 15 was different?

pg 6, line 236: How are you supporting that there is a delay in the deterioration of motor function? I don't think an RM-ANOVA can even accurately tell you that.

Figure 3: In the figure caption, there are different sample sizes for the grip strength and walking tests--why?

Discussion

pg 7, lines 250-252: based on above comments about results, I don't know that this opening summary is supported by the data.

pg 7, line 257: did the mice in your experiment receive any kind of chemotherapy?

pg 7, lines 261: in mice or in humans?

pg 7, lines 265-266: No--the p value was 0.167, therefore you did not find this.

Author Response

Point to point response letter to the Reviewers’ comments:

In the revised version of the manuscript (ms), corrections to the former ms have been highlighted in red to simplify the reviewing process.

Response to Reviewer #1: (reviewer’s comments are in italics)

We thank the Reviewer for the careful reading of our manuscript.

1) pg 1, line 29-30: though environmental enrichment may be well known to you, not all readers may be as familiar with it. Please at least list, if not summarize, the components other than voluntary physical exercise.

We have now summarized the principal components of environmental enrichment in the Introduction, acknowledging the proper references.

2) page 2, lines 49-51: Please make explicit if these statements apply only to animal models (I assume so since it ends with running wheels).

We thank the reviewer for allowing us to better clarify this point: we have now made it explicit in the text.

3) Similarly, throughout the manuscript, please make explicit when referenced studies dealt with animal vs human subjects. 

According to the reviewer’s suggestion we have now added those details in the text.

4) pg 2, lines 68-74: this belongs in methods; pg 2, lines 74-77: this belongs in discussion or conclusion. 

We have now removed those parts from the Introduction, according to reviewer’s advice.

5) Methods: pg 2, line 81: How many mice? 

We have now reported the number of mice used in total and for each analysis.

6)After reading the results, I am wondering why Mann-Whitney tests were mentioned in the statistical analysis subsection, since they weren't reported as used in the results. Were all data normally distributed with equal variances, thus not requiring Mann-Whitney tests? If so, why bother mentioning them in the methods? 

We apologize with the reviewer for the mistake. We have now removed the wrong sentence from the methods section.

7) Results: pg 4, lines 173-178: this is redundant of the methods.

We have now removed this part, according to reviewer’s advice.

8) pg 4, line 179: where is the data used for t-test analysis comparing runners and naive animals coming from? There's no mention of naive animals in this study. 

We added a part in Methods section explaining the number and the use of naive animals.

9) Figure 1. What is the yellow background section with many circles that are divided by right angles? This isn't explained in the figure caption. 

We have now added to the legend of figure 1 the explanation that the image cited by the reviewer was a schematic representation of the running wheel used in the work.

10) pg 5, line 210: delete "we found a trend for decreased tumor volumes;" there was no significance, therefore no effect. Using trend in this way is incorrect in statistics. Similarly, in Figure 2, delete "there is a tendency for decrease glioma size in running animals"--your statistics do not support that statement. 

We have now changed the sentences, as requested.

11) pg 6, lines 228: which effect is the reported p value of 0.033 referring to?

The reported p value is referred to treatment x time interaction.

12) pg 6, line 235: the omnibus F statistic from the RM-ANOVA would not have detected a single day’s change; what is the specific p value and what effect or extra test did you specifically run to detect that day 15 was different?

We apologize for the omission in the text. We performed the RM-ANOVA, followed by All Pairwise Multiple Comparison Procedures (Holm-Sidak post Hoc) and we found that the single p value at day 15 was 0.009 (Fig.3C).

13) pg 6, line 236: How are you supporting that there is a delay in the deterioration of motor function? I don’t think an RM-ANOVA can even accurately tell you that.

We thank the reviewer for asking us to clarify this point. Since RM-ANOVA evaluates time x treatment interaction, it is possible to appreciate that at day 15 running mice performed statistically significative better than the sedentary group, suggesting that physical exercise slowed down, in a transitory way, the deterioration of motor abilities in glioma-bearing animals.

14) Figure 3: In the figure caption, there are different sample sizes for the grip strength and walking tests--why? 

We reported a different sample size in fig 3B vs. fig 3D because we have identified some mice as outliers (ROUT, calculated with GraphPad Prism 8), thus they were removed.

15) Discussion pg 7, lines 250-252: based on above comments about results, I don't know that this opening summary is supported by the data. 

We have now changed the opening summary as suggested by the reviewer.

16) pg 7, line 257: did the mice in your experiment receive any kind of chemotherapy? 

We thank the reviewer for allow us to better clarify this point: our animals didn’t receive any chemotherapy; we have now specified it in the discussion.

17) pg 7, lines 265-266: No--the p value was 0.167, therefore you did not find this.

We have now changed this sentence, following the Reviewer suggestion.

Reviewer 2 Report

This study is reasonable and largely avoids overstating their findings which are not powered to be statistically robust. As has been demonstrated in other ways they show a modest improvement in motor testing in the context of physical exercise and some suggestive data that could indicate that in addition to retention of motor activity their could be a modest improvement in disease progression. While in both the conclusion and introduction they point to potential molecular mechanisms that could be contributing to this differential they provide no molecular evidence that these events are occurring in their animals which could have been done using ELISAs for changes in systemic IGF levels (for example) or by looking at the relevant signaling cascades in your tumor cohorts.   These studies would need to be repeated in larger cohorts with multiple models and appropriately powered in order to draw any definitive conclusions.  As such the claims in the abstract overstate the evidence provided in the figures.  In reviewing the effects seen on motor function may just relate to tumor size.  I would be curious to know if the (not significant difference in tumor volume at day 21, which may just result from normal variance of the system) could be used to normalize the observed differences in motor function.  That is can the motor function differences be ascribed simply to the difference in tumor size.     

Author Response

Point to point response letter to the Reviewers’ comments:

In the revised version of the manuscript (ms), corrections to the former ms have been highlighted in red to simplify the reviewing process.

Response to Reviewer #2: (reviewer’s comments are in italics)

We are grateful with the reviewer for a careful reading of our manuscript.

1) While in both the conclusion and introduction they point to potential molecular mechanisms that could be contributing to this differential they provide no molecular evidence that these events are occurring in their animals which could have been done using ELISAs for changes in systemic IGF levels (for example) or by looking at the relevant signaling cascades in your tumor cohorts.

We thank the reviewer for raising this interesting point. Unfortunately, we didn't have enough time to perform the experiments suggested: GL261 cells required 21 days to create the glioma mass and, due to editorial policies, we couldn't have enough time to perform ELISA assays to address this specific request. However, many papers have already proven that both IGF1 and BDNF levels increased after voluntary wheel running (Vivar et al. Curr Top Behav Neurosci. 2013; Baek J of Exercise Rehabilitation 2016; Kitamura et al., Neurosci Res 2003), and our central interest was understanding whether physical exercise might induce a general well-being on glioma-affected animals. Nonetheless, we had time to add more behavioral tests aimed at evaluating the overall healthiness of running and sedentary glioma-bearing animals (see Fig.4; Suppl Fig1; Suppl Fig2).

2) I would be curious to know if the (not significant difference in tumor volume at day 21, which may just result from normal variance of the system) could be used to normalize the observed differences in motor function.  That is can the motor function differences be ascribed simply to the difference in tumor size.

We are grateful with the reviewer for this valuable comment. We have performed a correlation between volume size and behavioral performances, but we didn't find any statistical proof that the motor function differences could be ascribed simply to the differences in tumor size.

3) As such the claims in the abstract overstate the evidence provided in the figures (tumor size).

We have now downstate the claims regarding the tumor size, as suggested by the reviewer.

Reviewer 3 Report

This article entitled “Voluntary physical exercise restrains tumor cell proliferation and ameliorates glioma-induced motor dysfunctions” by Elena Tantillo, et al. is a study of the effect of physical exercise on the tumor proliferation using a mouse model of high-grade glioma. The authors found that physical exercise reduced proliferation rates of tumors implanted in the motor cortex and delayed glioma-induced motor dysfunction. They concluded that voluntary physical exercise might represent a supportive intervention that complement existing neuro-oncologic therapies. The subject of this study will be of interest to readers of International Journal of Environmental Research and Public Health, however, I have the following concerns on the current form.

Comments

1. I would be afraid that the current form is lacking in certain data to conclude the effect of physical exercise against a proliferation of glioma cells. The authors present the potential data showing that physical exercise may restrain tumor cell proliferation, however, the additional experiments are necessary to confirm the results. The detailed mechanisms of antitumor effect should be clarified. Changes in immune system, oxidative stress, microcirculation, etc could be candidates causing anti-glioma effect of physical exercise. The authors are encouraged to add more experiments to enhance reliability of their results. 

Author Response

Point to point response letter to the Reviewers’ comments:

In the revised version of the manuscript (ms), corrections to the former ms have been highlighted in red to simplify the reviewing process.

Response to Reviewer #3: (reviewer’s comments are in italics)

We thank the reviewer for the careful reading of our manuscript.

The authors present the potential data showing that physical exercise may restrain tumor cell proliferation (...). The detailed mechanisms of antitumor effect should be clarified. Changes in immune system, oxidative stress, microcirculation, etc could be candidates causing anti-glioma effect of physical exercise. The authors are encouraged to add more experiments to enhance reliability of their results.

We thank the reviewer to point this out. We performed a well-established evaluation of proliferation using well-accepted and widely used markers (i.e. Ki67, BrdU; Venkatesh et al., 2015 Cell; Garofalo et al., 2015 Nat Comm) and we found a statistical decrease in tumor cell proliferation in running glioma-bearing animals. However, we weren't able to better dissect the mechanisms of antitumor effect of physical exercise because  we didn't have enough time to perform the experiments: GL261 cells required 21 days to create the glioma mass and, due to editorial policies, we couldn't have enough time to perform molecular assays to address this specific request. However, many papers have already proven that IGF1 and BDNF levels increased after voluntary wheel running (Vivar et al. Curr Top Behav Neurosci. 2013; Baek J of Exercise Rehabilitation 2016) and that physical exercise is capable of reducing oxidative stress (Radak et al., J Sport Health Sci 2013; Accattato et al., Plos ONE 2017; Simioni et al., Oncotarget 2018) and ameliorating microcirculation (Barnes and Corkery Brain Plast 2018). We have now added those studies in the discussion section. Nonetheless, our central interest was understanding whether physical exercise might induce a general well-being on glioma-affected animals and we had time to add more behavioural tests aimed at evaluating the overall healthiness of running and sedentary glioma-bearing animals (see Fig.4; Suppl Fig1; Suppl Fig2). 

Round 2

Reviewer 1 Report

Thank you for your thorough responses to all reviewer comments. I only found one minor piece in the revision that required further revision:

pg 8, line 286: Rewrite "We also performed a bunch of/three tests..." to be more clear and succinct

Author Response

We thank the Reviewer for the careful reading of our manuscript and for the suggestions. The mistake we made on page 6 has been now corrected.

Reviewer 2 Report

The authors have been non-responsive to the substance of the critique.  Their responses to the comments was that they did not have time, and/or did not see anything but they did not show any evidence to support these statements and made no substantive changes to the data to assuage the underlying concerns.   

For example in their response they write:

We are grateful with the reviewer for this valuable comment. We have performed a correlation between volume size and behavioral performances, but we didn't find any statistical proof that the motor function differences could be ascribed simply to the differences in tumor size.

Their title indicates that they find an effect on tumor cell proliferation however this claim is based on a "trend" and not as they say "statistical proof", so this response rings hollow as it sets a different standard for observations that support their hypothesis over the questions asked in the context of the review.  

Author Response

We thank the Reviewer for the careful reading of our manuscript.
Below, a point-by-point response to his/her criticisms.

1) "The authors have been non-responsive to the substance of the critique.  Their responses to the comments was that they did not have time (...) made no substantive changes to the data to assuage the underlying concerns."

We did collect blood and brain samples from the experimental groups in order to perform the molecular tests suggested, but the kit haven't arrived so far and we weren't able to add those analyses. Thus, in order to don't miss the deadline, we were forced to justify our hypothesis only by adding literatery references and behavioural tests (see fig 4, suppl fig 1 and suppl fig 2). However, we do understand that this represents a missing point.  

2) "Their title indicates that they find an effect on tumor cell proliferation however this claim is based on a "trend" and not as they say "statistical proof", so this response rings hollow as it sets a different standard for observations that support their hypothesis over the questions asked in the context of the review."  

We thank the reviewer for giving us the chance to better explain this point. We did find a statistical significant decrease in tumor proliferation on both used markers (i.e. BrdU and KI67; see Fig.1D-1F, p <0.001). On the contrary, we did not find any statistical differences between the two experimental groups regarding the volume of the tumor (Fig. 2). We have discussed this point, claiming that probably, despite the beneficial circulating factors due to physical exercise, tumoral growth is too strong and fast to be significantly impacted by a mild treatment as physical exercise. 

We would like to thank the reviewer for the attention and we look forward to hearing from them.

Reviewer 3 Report

The authors have failed to add experiments to enhance the reliability of their results because there was not enough time to do them. Instead of the additional experiments, the authors added literature review about potential molecular mechanisms of antitumor effect of physical exercise in Discussion. 

Author Response

We thank the Reviewer for their comprehension and willingness. We would like to specificy that we did collect blood and brain samples from the experimental groups in order to perform the molecular tests suggested by reviewer 2 and 3, but the kit haven't arrived so far. Thus, we were forced to justify our hypothesis only by adding literatery references and behavioural tests (see fig 4, suppl fig 1 and suppl fig 2).